# Capsule networks as recurrent models of grouping and segmentation

**Adrien Doerig**[1]☯*, **Lynn Schmittwilken**[1,2]☯, **Bilge Sayim**[3,4], **Mauro Manassi**[5], **Michael H. Herzog**[1]

**1** Laboratory of Psychophysics, Brain Mind Institute, École Polytechnique Fédérale de Lausanne (EPFL), Lausanne, Switzerland, **2** Dept. Computational Psychology, Institute of Software Engineering and Theoretical Computer Science, Technische Universität Berlin, Berlin, Germany, **3** Institute of Psychology, University of Bern, Bern, Switzerland, **4** Univ. Lille, CNRS, UMR 9193—SCALab—Sciences Cognitives et Sciences Affectives, F-59000 Lille, France, **5** School of Psychology, University of Aberdeen, Scotland, United Kingdom

☯ These authors contributed equally to this work.
* adrien.doerig@gmail.com

## Abstract

Classically, visual processing is described as a cascade of local feedforward computations. Feedforward Convolutional Neural Networks (ffCNNs) have shown how powerful such models can be. However, using visual crowding as a well-controlled challenge, we previously showed that no classic model of vision, including ffCNNs, can explain human global shape processing. Here, we show that Capsule Neural Networks (CapsNets), combining ffCNNs with recurrent grouping and segmentation, solve this challenge. We also show that ffCNNs and standard recurrent CNNs do not, suggesting that the grouping and segmentation capabilities of CapsNets are crucial. Furthermore, we provide psychophysical evidence that grouping and segmentation are implemented recurrently in humans, and show that CapsNets reproduce these results well. We discuss why recurrence seems needed to implement grouping and segmentation efficiently. Together, we provide mutually reinforcing psychophysical and computational evidence that a recurrent grouping and segmentation process is essential to understand the visual system and create better models that harness global shape computations.

**Data Availability Statement:** The human data for experiment 2 and the full code to reproduce all our results are available here: https://github.com/adriendoerig/Capsule-networks-as-recurrent-models-of-grouping-and-segmentation.

## Author summary

Feedforward Convolutional Neural Networks (ffCNNs) have revolutionized computer vision and are deeply transforming neuroscience. However, ffCNNs only roughly mimic human vision. There is a rapidly expanding body of literature investigating differences between humans and ffCNNs. Several findings suggest that, unlike humans, ffCNNs rely mostly on local visual features. Furthermore, ffCNNs lack recurrent connections, which abound in the brain. Here, we use visual crowding, a well-known psychophysical phenomenon, to investigate recurrent computations in global shape processing. Previously, we showed that no model based on the classic feedforward framework of vision can explain global effects in crowding. Here, we show that Capsule Neural Networks (CapsNets), combining ffCNNs with recurrent grouping and segmentation, solve this challenge.

**Funding:** AD was supported by the Swiss National Science Foundation grant n.176153 "Basics of visual processing: from elements to figures". The funders had no role in study design, data collection and analysis, decision to publish, or preparation of the manuscript.

**Competing interests:** The authors have declared that no competing interests exist.

ffCNNs and recurrent CNNs with lateral and top-down recurrent connections do not, suggesting that grouping and segmentation are crucial for human-like global computations. Based on these results, we hypothesize that one computational function of recurrence is to efficiently implement grouping and segmentation. We provide psychophysical evidence that, indeed, grouping and segmentation is based on time consuming recurrent processes in the human brain. CapsNets reproduce these results too. Together, we provide mutually reinforcing computational and psychophysical evidence that a recurrent grouping and segmentation process is essential to understand the visual system and create better models that harness global shape computations.

## Introduction

The visual system is often seen as a hierarchy of local feedforward computations [1], going back to the seminal work of Hubel and Wiesel [2]. Low-level neurons detect basic features, such as edges. Pooling these outputs, higher-level neurons detect higher-level features such as corners, shapes, and ultimately objects. Feedforward Convolutional Neural Networks (ffCNNs) embody this classic framework of vision and have shown how powerful it can be [e.g., 3–6]. However, ffCNNs only roughly mimic human vision as a large body of literature shows. For example, ffCNNs lack the abundant recurrent processing of humans [7,8], perform differently than humans in crucial psychophysical tasks [9,10], and can be easily misled [11–13]. An important point of discussion concerns global visual processing. It was suggested that ffCNNs mainly focus on local, texture-like features, while humans harness global shape computations ([9,13–17]; but see [18]). In this context, it was shown that changing local features of an object, such as its texture or edges, leads ffCNNs to misclassify [13,14], while humans can still easily classify the object based on its global shape.

There are no widely accepted diagnostic tools to specifically characterize global computations in neural networks. Models are usually compared on computer vision benchmarks, such as ImageNet [19], or with neural responses in the visual system [20,21]. One drawback of these approaches is that the datasets are hard to control. Psychophysical results can be used to fill this gap and create well-controlled challenges for visual models, tailored to target specific aspects of vision [22]. Here, we use visual crowding to specifically target global shape computations in humans and artificial neural networks.

In crowding, objects that are easy to identify in isolation appear as jumbled and indistinct when clutter is added [9,23–28]. Consider the following example: Fig 1A shows a vernier target, i.e., two vertical lines separated by a horizontal offset. When the vernier is presented alone, observers easily discriminate the offset direction. When a flanking square surrounds the target, performance drops, i.e., there is strong crowding [29,30]. Surprisingly, *adding* more flankers can *reduce* crowding strongly, depending on the spatial configuration of the flankers (Fig 1B; [28]). Hence, the *global* configuration of visual elements across large regions of the visual field influences perception of the small vernier target. This global *un*crowding effect occurs for a wide range of stimuli in vision, including foveal and peripheral vision, audition, and haptics [31–37]. The ubiquity of (un)crowding in perception is not surprising, since visual elements are rarely seen in isolation. Hence, any perceptual system needs to cope with crowding to isolate important information from clutter.

Previously, we have shown that these global effects of crowding *cannot* be explained by models based on the classic framework of vision, including ffCNNs [9,17,38]. Here, we propose a new framework to understand these global effects. We show that Capsule Neural Networks (CapsNets;

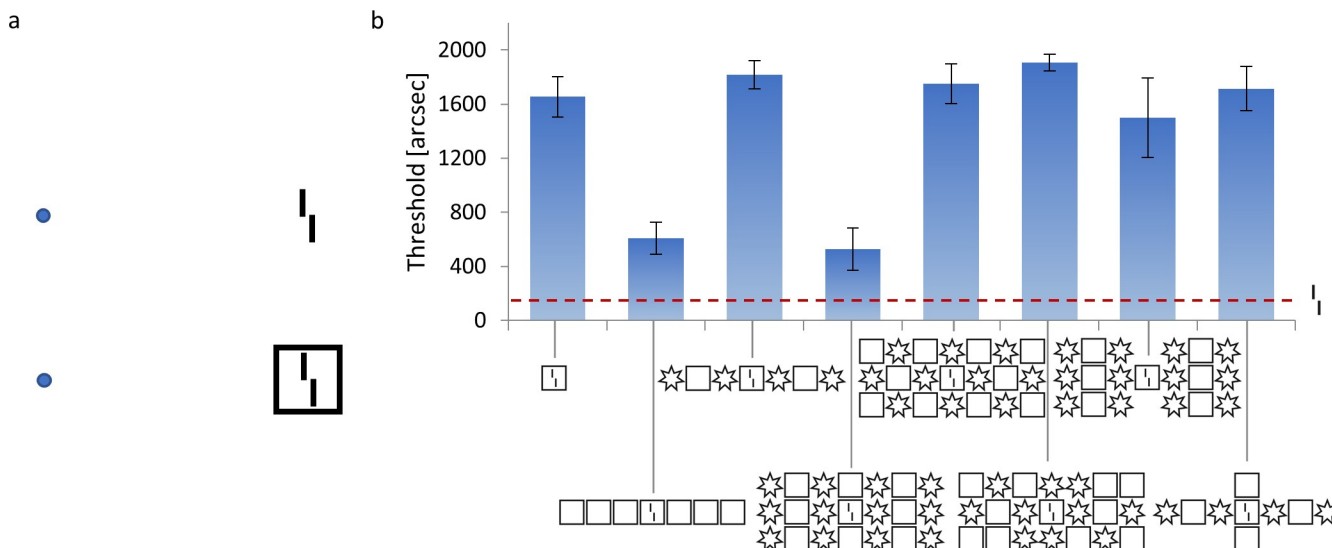

**Fig 1. (a) Crowding:** The perception of visual elements deteriorates in clutter, an effect called crowding. When fixating on the blue dots, the vernier (i.e., two vertical bars with a horizontal offset) becomes harder to perceive when it is flanked by a square. **(b) Uncrowding:** Results of a psychophysical experiment in which stimuli are presented in the visual periphery. The y-axis shows the minimal offset size at which observers can report the offset direction with 75% accuracy (i.e., lower values indicate better performance). The offset direction of a vernier in isolation can be easily reported (dashed red line). When a flanking square surrounds the vernier, performance deteriorates (crowding). When more squares are added, performance recovers (uncrowding). Critically, the uncrowding effect depends on the global stimulus configuration. For example, if some squares are replaced by stars, performance deteriorates again (3rd bar; [28]).

[39]), augmenting ffCNNs with a recurrent grouping and segmentation process, can explain these complex global (un)crowding results in a natural manner. Two processing regimes can occur in CapsNets: a fast feedforward pass that is able to quickly process information, and a time-consuming recurrent regime to compute in-depth global grouping and segmentation. We will show psychophysical evidence that the human visual system indeed harnesses recurrent processing for efficient grouping and segmentation, and that CapsNets naturally reproduce this behavior as well. Together, our results suggest that a time-consuming recurrent grouping and segmentation process is crucial for global shape computations in both humans and artificial neural networks.

## Results

### Experiment 1: Crowding and Uncrowding Naturally Occur in CapsNets

In CapsNets, early convolutional layers extract basic visual features. Recurrent processing combines these features into groups and segments objects by a process called *routing by agreement* (in most implementations of CapsNets, including ours and [39], the iterative routing by agreement process is not explicitly implemented as a "standard" recurrent neural network processing sequences of inputs online. Instead, there is an iterative algorithmic loop (see [39] for the algorithm), which is equivalent to recurrent processing). The entire network is trained end-to-end through backpropagation. *Capsules* are groups of neurons representing visual features and are crucial for the routing by agreement process. Low-level capsules iteratively predict the activity of high-level capsules in a recurrent loop. If the predictions agree, the corresponding high-level capsule is activated. For example, if a low-level capsule detects a rectangle, and the capsule above detects a triangle, as shown in Fig 2A, they agree that the higher-level object should be a house and, therefore, the corresponding high-level capsule is activated (Fig 2A). This process allows CapsNets to group and segment objects (Fig 2B). Because of these capabilities, we hypothesized that CapsNets are able to reproduce human-like (un)crowding in a visual crowding experiment.

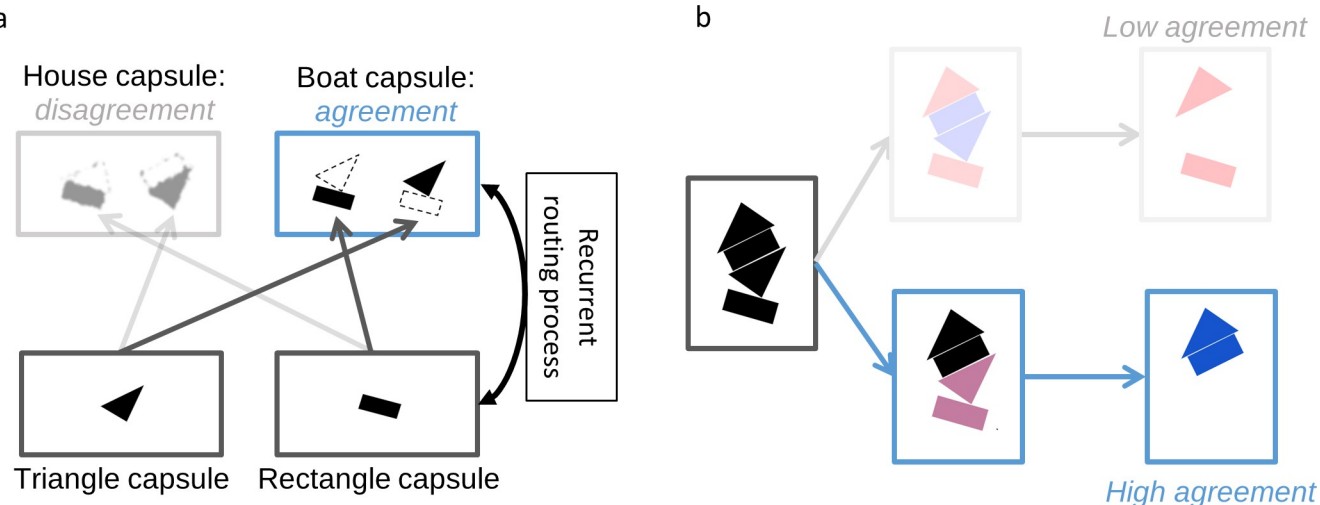

**Fig 2. (a) Routing by agreement in CapsNets:** A capsule is a group of neurons whose activity vector represents an entity as well as its instantiation parameters (such as position, orientation, color etc.). In this example, lower-level capsules detect triangles and rectangles as well as their orientations. Higher-level capsules detect combinations of these shapes. Here, the triangle capsule detects a tilted triangle and the rectangle capsule detects a tilted rectangle. Then, each capsule predicts the representation of the higher-level capsule. Based on the orientation of its own representation, the triangle capsule predicts an upside-down house and a tilted boat. Similarly, the rectangle capsule predicts a tilted house and a tilted boat. In CapsNets, information between layers gets propagated recurrently in order to maximize their agreement. In this case, capsules agree on the orientation of the boat but disagree on the orientation of the house. Hence, the routing by agreement process suppresses activity in the house capsule and boosts activity in the boat capsule. **(b) Grouping and segmentation in CapsNets:** The recurrent routing by agreement process endows CapsNets with natural grouping and segmentation capabilities. Here, an ambiguous stimulus is presented, which can either be seen as an upside-down house (top) or a house and a boat (bottom). The upside-down house interpretation leaves parts of the image unexplained and this causes disagreement. Routing by agreement will boost the bottom interpretation, because it leads to most agreement in the network. In order to do so, the individual features get grouped into a house and a boat and segmented into the corresponding higher-level capsules.

We trained CapsNets with two convolutional layers followed by two capsule layers to recognize greyscale images of vernier targets and groups of identical shapes (see Methods). During training, either a vernier or a group of identical shapes was presented. The network had to simultaneously classify the shape type, the number of shapes in the group, and the vernier offset direction. Importantly, verniers and shapes were never presented together during training. Therefore, the network was never exposed to (un)crowding stimuli during training.

During testing, we evaluated the vernier offset discrimination of the networks when presented with (un)crowding stimuli. The CapsNets reproduce both crowding and uncrowding as in psychophysical experiments (Fig 3A): presenting the vernier within a single flanker deteriorated offset discrimination performance (crowding). Adding more identical flankers recovered performance (uncrowding). Adding configurations of alternating different flankers did not recover performance (crowding), similarly to human vision. Small changes in the network hyperparameters, loss terms or stimulus characteristics did not affect these results (Fig A in S1 Appendix). As a control condition, we checked that when the vernier target is presented outside the flanker configuration, rather than inside, there was no performance drop (S3 Appendix). Hence, the performance drop in crowded conditions was not merely due to the simultaneous presence of the target and flanking shape in the stimulus, but really comes from crowding between the vernier and flankers.

Because each feature is represented in its own capsule, we can decode how each feature of a stimulus is represented in its capsule to gain important insights about the underlying computations. This is an advantage compared to ffCNNs, in which it is unclear which neurons represent which feature. Reconstructing the input images based on the network's output (see Methods) shows that the difference between crowding and uncrowding comes from grouping and segmentation (Fig 3B). More specifically, crowding occurs when the target and flankers

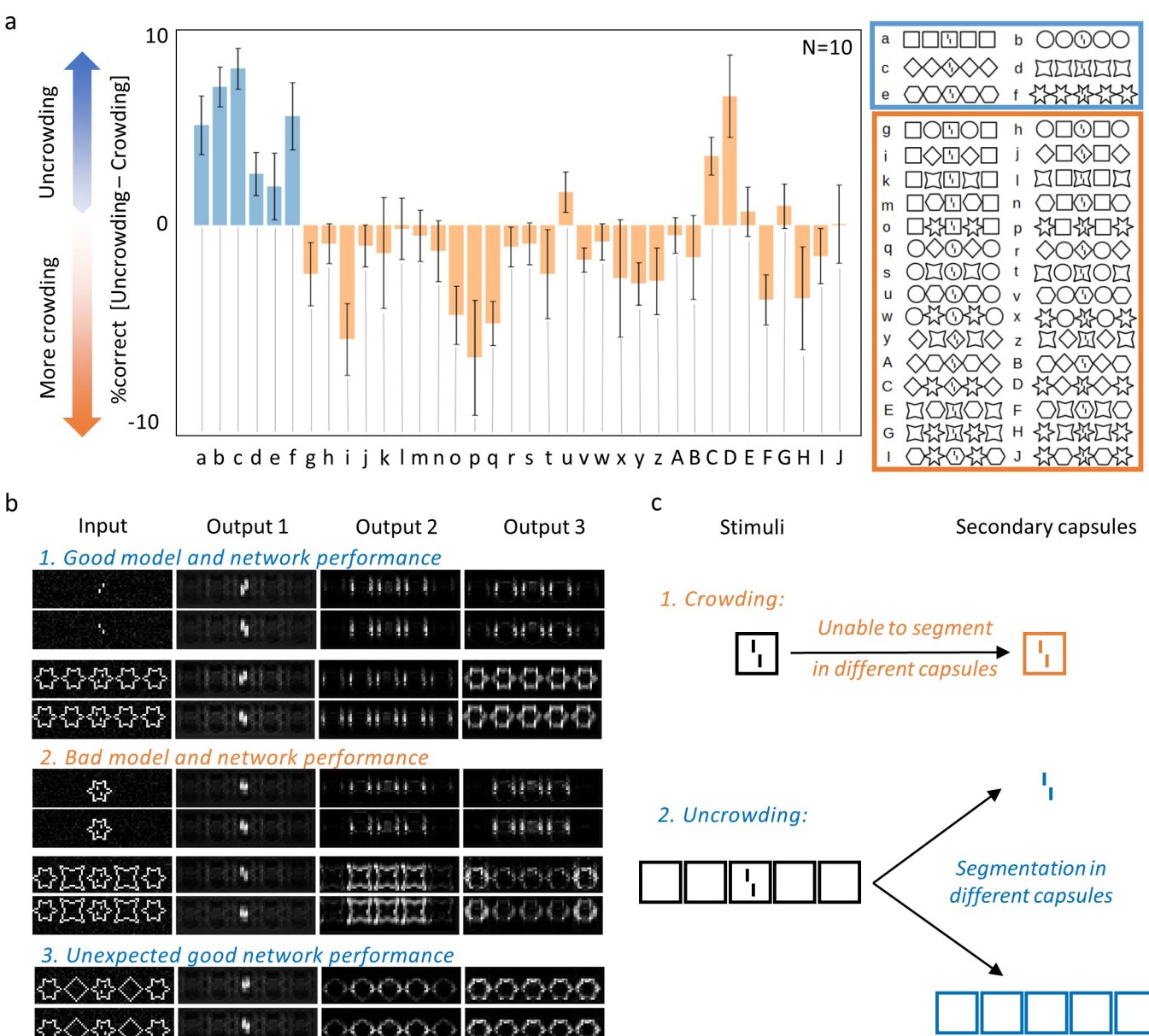

**Fig 3. a. CapsNets explain both crowding and uncrowding:** The x-axis shows the configurations that were used to test (un)crowding. We used 6 different flanker shape types and tested all configurations with 5 identical or alternating shapes (e.g., 5 squares, 5 circles, circle-square-circle-square-circle, etc; see Methods). Performance is shown on the y-axis as %correct for the whole configuration *minus* %correct for the central flanker alone. For example, in column *a*, vernier offset discrimination is better with 5 square flankers than with 1 square flanker. We trained N = 10 networks to match the typical number of observers in human experiments [28,40]. Error bars indicate the standard error across all networks. The blue bars represent configurations for which *un*crowding is expected. Values larger than 0 are in accordance with human data. Orange bars represent configurations for which crowding is expected. Values smaller than or equal to 0 are in accordance with human data. **b. Reconstructions:** We reconstructed the input images based on the secondary capsules' activities (see Methods). The reconstructions from the three most active secondary capsules are shown. When the vernier is presented alone (1. top), the reconstructions are good. When a single flanker is added (panel 2 top), the vernier and flanker interfere because the vernier is not well segmented from the flanker, and reconstructions deteriorate (crowding). Sometimes, the vernier is reconstructed with an incorrect offset. When identical flankers are added (panel 1 bottom), both the vernier and the flanker reconstructions recover, because the vernier is well segmented from the flankers (uncrowding). With different flankers (2. bottom), the segmentation is unsuccessful and reconstructions deteriorate again (crowding). Sometimes, the vernier is reconstructed with the incorrect offset. Interestingly, the reconstructions in case of "unexpected" uncrowding (i.e., the networks show uncrowding contrary to humans) often resemble the ones in case of "normal" uncrowding: one flanker type is well segmented and reconstructed (compare panel 3 with panels 1&2 bottom). This suggest that the network was unable to differentiate between diamonds and stars, and treated this configuration as a large group of stars. **c. Segmentation and (un)crowding in CapsNets:** If CapsNets can segment the vernier target away from the flankers during the recurrent routing by agreement process, uncrowding occurs. Segmentation is difficult when a single flanker surrounds the target because capsules disagree about what is shown at this location. In the case of configurations that the network has learnt to group, many primary capsules agree about the presence of a group of shapes. Therefore, they can easily be segmented away from the vernier.

cannot be segmented and are therefore routed to the same capsule. In this case, they interfere because a single capsule cannot represent well two objects simultaneously due to limited neural resources. This mechanism is similar to pooling: information about the target is pooled with information about the flankers, leading to poorer representations. However, if the flankers are segmented away and represented in a different capsule, the target is released from the flankers' deleterious effects and *un*crowding occurs (Fig 3C). This segmentation can only occur if the network has learnt to group the flankers into a single higher-level object represented in a different capsule than the vernier. Segmentation is facilitated when more flankers are added because more capsules agree on the presence of the flanker group.

Alternating configurations of different flankers, as in the third configuration of Fig 1B, usually do not lead to uncrowding in humans [28]. In some rare cases, our CapsNets produced uncrowding with such configurations (e.g. configurations u, C and D in Fig 3A). Reconstructions suggest that in these cases the network could not differentiate between the different shapes of the flankers (e.g. between diamonds and stars). Therefore, the flankers formed a group for the network and were segmented away from the target (Fig 3B). This further reinforces the notion that grouping and segmentation differentiate crowding from uncrowding: whenever the network reaches the conclusion that the flankers form a group, segmentation is facilitated. When this happens, the vernier and flankers are represented in different capsules and therefore do not interfere, leading to good performance.

In previous work, we have shown that pretrained ffCNNs cannot explain uncrowding [17], even if they are biased towards global shape processing [13]. Currently, CapsNets cannot be trained on large-scale tasks such as ImageNet because routing by agreement is computationally too expensive. Therefore, we took a different approach here. As explained above, we trained our CapsNets to recognize groups of shapes and verniers and asked how they would generalize from shapes presented in isolation to crowded shapes. To make sure that CapsNets explain global (un)crowding due to their *architecture* focusing on grouping and segmentation and not merely due to a difference in the *training regime*, we conducted three further experiments. We investigated how vernier discrimination performance changes when the capsule layers are replaced by other architectures, keeping the number of neurons constant.

First, we replaced the capsule layers by a fully connected feedforward layer, yielding a standard ffCNN with three convolutional layers and a fully connected layer. We trained and tested this architecture in the same way as the CapsNets. The results clearly show that there is no uncrowding (Fig 4A): ffCNNs do not reproduce human-like global computations with this procedure.

Second, we added lateral recurrent connections to the fully connected layer of the ffCNN, yielding a network with three convolutional layers followed by a fully connected recurrent layer. We used the same number of recurrent iterations as for the routing by agreement in the CapsNets, and trained the network the same way as before. There is no uncrowding with this architecture either (Fig 4B).

Lastly, we added top-down recurrent connections feeding back from the final fully connected layer of the ffCNN to the layer below, yielding a network with three convolutional layers followed by a fully connected layer that fed back into the previous layer (again with the same number of recurrent iterations as iterations of routing by agreement in the CapsNets). After the same training, this architecture did not produce any uncrowding either (Fig 4C).

The absence of uncrowding in ffCNNs and recurrent CNNs with lateral or top-down recurrent connections suggests that the *architecture* of CapsNets, and not our *training regime* explains why (un)crowding is reproduced. Furthermore, recurrent processing by itself is not sufficient to produce (un)crowding. The grouping and segmentation performed by routing by agreement seems crucial.

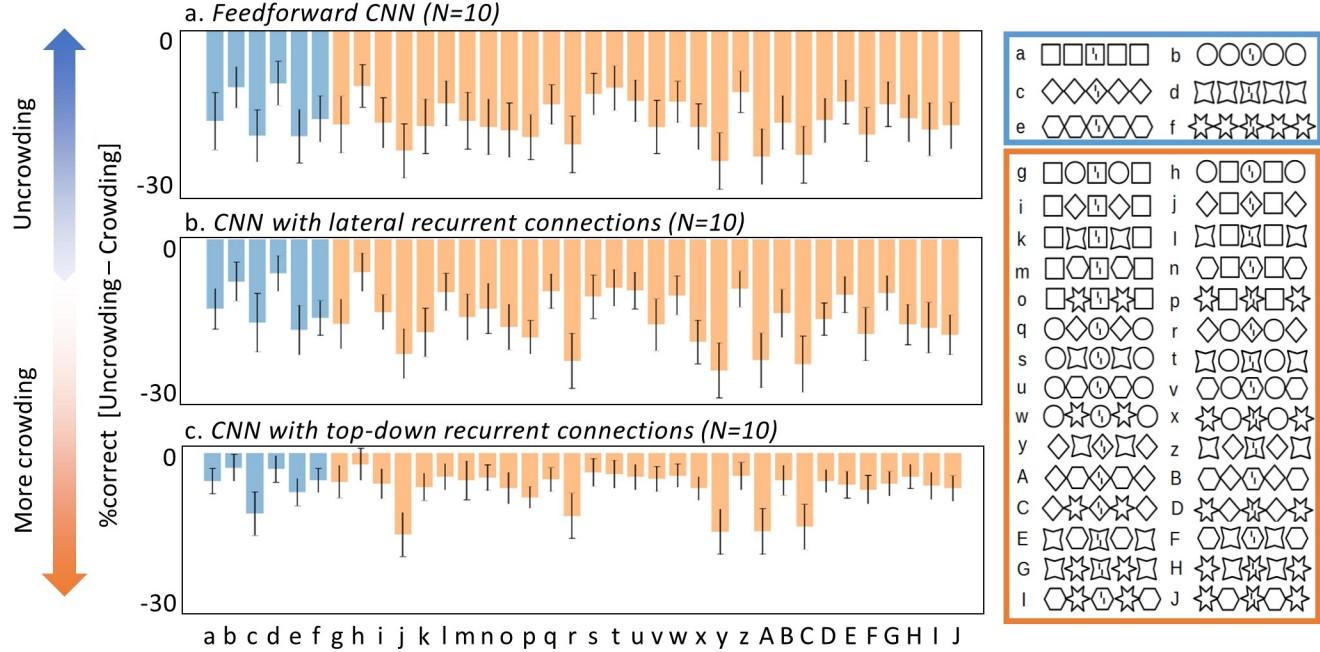

**Fig 4. Other network architectures do not explain uncrowding.** To verify that the ability of CapsNets to explain uncrowding is due to their architecture and not merely to the way they are trained, we replaced the recurrent routing by agreement between capsules by three different alternative architectures: a feedforward fully connected layer (yielding a classic ffCNN, a), a fully connected layer with lateral recurrent connections (b) and a fully connected layer with top-down recurrent connections to the layer below (c). The x-axes represent the (un)crowding configurations used during testing (shown on the right). The y-axes show the vernier discrimination performance as %correct for the whole configuration minus %correct for the central flanker alone as in Fig 3A. None of the ffCNN or recurrent CNN architectures we tested can produce uncrowding (compare with the CapsNet results in Fig 3A).

## Experiment 2: The role of recurrent processing

Processing in CapsNets starts with a feedforward sweep which is followed by a time-consuming recurrent routing by agreement process to refine grouping and segmentation. We hypothesized that the human brain may use a similar strategy to efficiently implement grouping and segmentation. To test this hypothesis, we psychophysically investigated the temporal dynamics of (un)crowding in humans.

For this, we performed a psychophysical crowding experiment with a vernier target flanked by either two lines or two cuboids (see Methods; Fig 5). The stimuli were displayed for varying durations from 20 to 640ms. The observers had to report the vernier offset direction. For short stimulus durations, crowding occurred for both flanker types, i.e., vernier offset thresholds were significantly larger in both the lines condition and cuboids conditions compared to the vernier alone condition (lines: p = 0.0017, cuboids: p = 0.0013, 2-tailed one-sample t-tests).

To quantify how performance changed with increasing stimulus duration, we fitted a line $y = ax+b$ to the data for each subject, and compared the slopes $a$ between the lines condition and the cuboids condition. Discrimination performance in the cuboids condition improved significantly more with increasing stimulus duration than performance in the lines condition (p = 0.003, 2-tailed 2-sample t-test). Previous results have suggested that crowding varies little with stimulus duration ([41]; but see [42,43]). As shown in Fig 5, this may be the case for the lines condition of our experiment, but *not* for the cuboids condition, where performance drastically improves with stimulus duration. This difference in performance between the lines and cuboids conditions cannot be explained by local mechanisms such as lateral inhibition [29,44] or pooling [45–47], since the inner vertical lines are the same for the line and cuboid flankers.

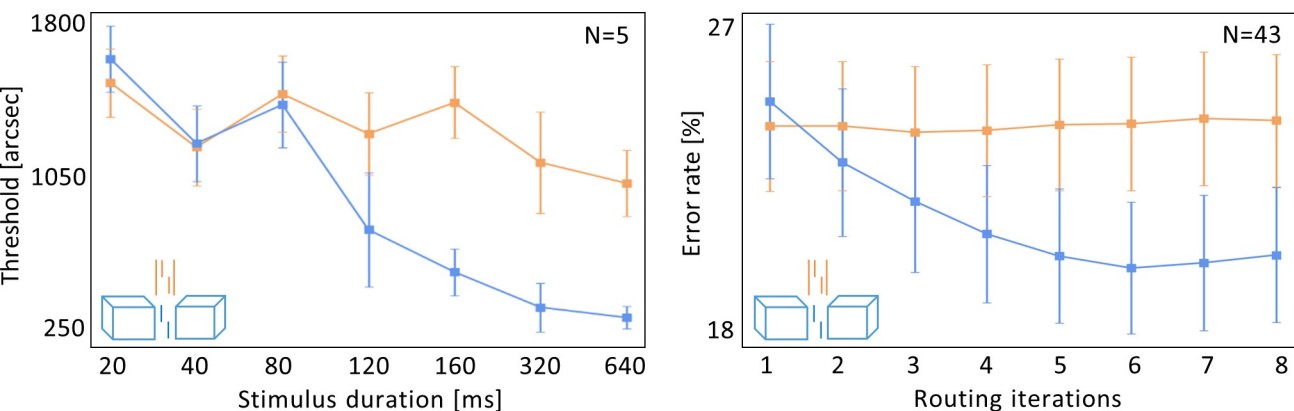

**Fig 5. Temporal dynamics of uncrowding:** *Left*: *Human data*. The x-axis shows the stimulus durations and the y-axis shows the corresponding thresholds (i.e., lower values indicate better performance). Error bars indicate the standard error. For cuboid flankers, crowding occurs up to 100ms of stimulus presentation, but performance gradually improves with increasing stimulus duration (i.e., uncrowding occurs with time, blue). In contrast, there is no uncrowding with increasing stimulus duration for the line flankers (orange). This suggests that the cuboids are segmented from the vernier through time-consuming recurrent processing. In contrast, the line flankers are strongly grouped with the vernier and cannot be segmented at all. *Right*: *Model data*. CapsNets can explain these results by varying the number of recurrent routing by agreement iterations. The x-axis shows the number of recurrent routing iterations and the y-axis shows the error rates for the vernier discrimination performance (i.e., lower values indicate better performance). Error bars indicate the standard error across all N = 43 trained networks (see Methods). CapsNets reproduce the results of the human psychophysical experiment: discrimination errors decrease with an increasing number of recurrent routing iterations, but only in the cuboids condition. This suggests that recurrent processing helps to compute and segment the cuboids, but the lines are too strongly grouped with the vernier to be segmented. Hence, they do not benefit from the recurrent segmentation process. Note that the blue and orange stimulus colors are for illustration only. All stimuli were presented on the same oscilloscope and had the same color.

Crucially, uncrowding occurred for the cuboid flankers only when stimulus durations were sufficiently long (Fig 4). We propose that these results reflect the time-consuming recurrent computations needed to segment the cuboid flankers away from the target. Performance does not improve as much with the line flankers, because they are too strongly grouped with the vernier target, so recurrent processing cannot segment them away.

To investigate if CapsNets reproduce these results, we trained CapsNets with the same architecture as in experiment 1 to discriminate vernier offsets, and to recognize lines, cuboids and scrambled cuboids (see Methods). The scrambled cuboids were included to prevent the network from classifying lines vs. cuboids simply based on the number of pixels in the image. As in experiment 1, each training sample contained only one of the shape types. We used 8 routing by agreement iterations during training. As in experiment 1, verniers and flankers were never presented together during training (i.e., there were no (un)crowding stimuli).

After training, we tested the networks on (un)crowding stimuli, changing the number recurrent routing by agreement iterations from one (leading to a purely feedforward regime) to 8 iterations (a highly recurrent regime; Fig 5). We found that CapsNets naturally explain the human results. Using the same statistical analysis as for the psychophysical experiment, we found that discrimination performance improves significantly more with increasing routing iterations in the cuboids condition as compared to the lines condition (p = 0.002; 2-tailed 2-sample t-test). As shown in Fig 5, additional processing time only has a small effect on performance for the line stimuli. In contrast, with more iterations the cuboids are better segmented from the target, and performance improves strongly. These results were not affected by small changes in network hyperparameters, loss terms or by changing stimulus details such as changing the cuboids from opaque to transparent (Fig B in S1 Appendix). We did not compare these results with the ffCNN and recurrent networks used in experiment 1, because these networks produced no uncrowding at all.

These findings are explained by the recurrent routing by agreement process. With cuboids, capsules across an extended spatial region need to agree about the presence of a cuboid, which is then segmented into its own capsule. This complex process requires several recurrent iterations of the routing by agreement process. On the other hand, the lines are much more strongly grouped with the vernier, so further iterations of routing by agreement achieve worse segmentation and, hence, cannot improve performance as much.

## Discussion

Our results provide strong evidence that time-consuming recurrent grouping and segmentation is crucial for shape-level computations in both humans and artificial neural networks. We used (un)crowding as a psychophysical probe to investigate how the brain flexibly forms object representations. These results specifically target global, shape-level and time-consuming recurrent computations and constitute a well-controlled and difficult challenge for neural networks.

It is well known that humans can solve a number of visual tasks very quickly, presumably in a single feedforward pass of neural activity [48]. ffCNNs are good models of this kind of visual processing [20,21,49]. However, many studies have shown that neural activities are not determined by the feedforward sweep alone, and recurrent activity affords a distinct processing regime to perform more in-depth and time-consuming computations [7,8,50–53]. Similarly, CapsNets naturally include both a fast feedforward and a time-consuming recurrent regime. When a single routing by agreement iteration is used, CapsNets are rapid feedforward networks that can accomplish many tasks, such as vernier discrimination or recognizing simple shape types (e.g. circles vs. squares). With more routing iterations, a recurrent processing regime arises and complex global shape effects emerge, such as segmenting the cuboids in experiment 2. We showed how the transition from feedforward to recurrent processing in CapsNets explains psychophysical results about temporal dynamics of (un)crowding.

Recurrent activity offers several advantages. First, although feedforward networks can in principle implement any function [54], recurrent networks can implement certain functions more efficiently. Flexible grouping and segmentation is exactly the kind of function that may benefit from recurrent computations (see also [55]). For example, to determine which local elements should be grouped into a global object, it helps to compute the global object first. This information can then be fed back to influence how each local element is processed. For example, to model (un)crowding, it helps to compute the global configuration of flankers first to determine how to process the vernier. Should it be grouped with the flankers (crowding) or not (uncrowding)? In CapsNets, the first feedforward sweep of activity provides an initial guess about which global objects are present (e.g., large cuboids). At this stage, as shown in experiment 2, information about the vernier interferes with information about the cuboids (crowding). Then, recurrent processing helps to route the relevant information about the cuboids and the vernier to the corresponding capsules (uncrowding). Without recurrence, it is difficult to rescue the vernier information once it has been crowded.

Second, although any network architecture can implement any computation in principle (given enough neurons), they differ in the way they *generalize* to previously unseen stimuli. Hence, recurrent grouping and segmentation architectures influence what is learnt from the training data. Here, we have shown that only CapsNets, but not ffCNN or CNNs augmented with recurrent lateral or top-down connections, produce uncrowding when trained identically to recognize groups of shapes and verniers. In general, ffCNNs tend to generalize poorly (review: [56]). Using different architectures to improve how current systems generalize is a promising avenue of research. In this respect, we have shown that CapsNets generalize more similarly to humans than ffCNNs and standard recurrent networks in the context of global (un)crowding. Relatedly,

CapsNets may also be more resilient to adversarial attacks [57,58], generalize better to new viewpoints [59], and outperform ffCNNs on difficult visual tasks with scarce training data such as tumor detection [60], also suggesting potentially more human-like generalization than ffCNNs.

One limitation in our experiments is that we explicitly taught the CapsNets which configurations to group together by selecting which groups of shapes were present during training (e.g., only groups of identical shapes in experiment 1). Effectively, this gave the network adequate priors to produce uncrowding with the appropriate configurations (i.e., only identical, but not different flankers). Hence, our results show that, given adequate priors, CapsNets explain uncrowding. We have shown that ffCNNs and CNNs with lateral or top-down recurrent connections do *not* produce uncrowding, *even* when they are trained identically on groups of identical shapes and successfully learn on the training data, comparably to the CapsNets (furthermore, we showed previously that ffCNNs trained on large datasets, which are often used as general models of vision, do not show uncrowding either; [17]). This shows that merely training networks on groups of identical shapes is not sufficient to explain uncrowding. It is the recurrent segmentation in CapsNets that is crucial. Humans do not start from zero and therefore do not need to be trained in order to perform crowding tasks. The human brain is shaped through evolution and learning to group elements in a useful way to solve the tasks it faces. As mentioned, (un)crowding can be seen as a probe into this grouping strategy. Hence, we expect that training CapsNets on more naturalistic tasks such as ImageNet may lead to grouping strategies similar to humans and may therefore naturally equip the networks with priors that explain (un)crowding results. At the moment, however, CapsNets have not been trained on such difficult tasks because the routing by agreement algorithm is computationally too expensive.

The approach we took in this contribution goes beyond the standard use of deep learning in neuroscience, in which large ffCNN are fitted on large datasets to model behavior. To our knowledge, there is little research on the *specific* kinds of computations that ffCNNs *cannot* do and, importantly, *why they cannot do it*. Here, we have shown that a very specific kind of recurrent computations is needed to explain global processing: recurrent grouping and segmentation. Furthermore, experiment 2 goes beyond the usual static image recognition measures by taking time-extended computations into account. There are currently very few attempts to explain the dynamics of human neural computations using deep learning (with the notable exception of [8]). This may be because most researchers use ffCNNs, which are ill-suited to address such dynamic computations. Contrasting different non-standard deep learning architectures to ask specific neuroscientific questions about well-defined tasks can lead to more precise ways of using deep networks in neuroscience. We think such approaches will be important in the future.

Recurrent networks are harder to train than feedforward systems, which explains the dominance of the latter during these early days of deep learning. However, despite this hurdle, recurrent networks are emerging to address the limitations of ffCNNs as models of the visual system [8,50,52,53,61,62]. Although there is consensus that recurrence is important for brain computations, it is currently unclear which functions exactly are implemented recurrently, and how they are implemented. Our results suggest that one important role of recurrence is global shape-level computation through grouping and segmentation. We had previously suggested another recurrent segmentation network, hard-wired to explain uncrowding [63]. However, CapsNets, bringing together recurrent grouping and segmentation with the power of deep learning, are much more flexible and can be trained to solve any task. Linsley et al. [53] proposed another recurrent deep neural network for grouping and segmentation, and there are other possibilities too [64,65]. We do not suggest that CapsNets are the only implementation of grouping and segmentation. We only suggest that grouping and segmentation is important and CapsNets are one potential way in which grouping could be solved. Further work is needed to show how the brain implements it. Regardless of the specific implementation, a focus on recurrent grouping and

segmentation offers a fresh way of conceptualizing global visual processing and has important implications far beyond crowding, for neuroscience and computational science in general. Harnessing the power and flexibility offered by deep learning approaches beyond classic ffCNNs will be crucial to understand and model these complex processes. We suggest that important new avenues of research will open up when complex deep networks such as CapsNets can be scaled to deal with large complex datasets, and this contribution is an example of this promising approach.

In conclusion, our results provide mutually reinforcing modelling and psychophysical evidence that time-consuming, recurrent grouping and segmentation plays a crucial role for global shape computations in humans and machines.

## Methods

The code to reproduce our results is available at https://github.com/adriendoerig/Capsule-networks-as-recurrent-models-of-grouping-and-segmentation. All models were implemented in Python 3.6, using the high-level estimator API of Tensorflow 1.10.0. Computations were run on a GPU (NVIDIA GeForce GTX 1070).

We used the same basic network architecture in all experiments, motivated by the following rationale (Fig 6A). After training, ideally, primary capsules detect the individual shapes present in the input image, and the secondary capsules group and segment these shapes through recurrent routing by agreement. The network can only group shapes together because it was taught during training that specific shapes form a group. To match this rationale, we set the primary capsules' receptive field sizes to roughly match the size of one shape.

Early feature extraction was implemented with two convolutional layers without padding, each followed by an ELU non-linearity. These convolutional layers were followed by two capsule layers using routing by agreement. The primary capsules were implemented by a third convolution layer, reshaped into $m$ primary capsule types outputting $n$-dimensional activation vectors (this is the standard way to implement primary capsules). Finally, the number of secondary capsule types was equal to the number of different shapes used as input. Different decoders were implemented to measure different loss terms (Eqs 1–7). Each decoder used the secondary capsules as inputs. The networks were trained end-to-end through backpropagation. For training, we used an Adam optimizer with a batch size of 48 and a learning rate of 0.0004. To this learning rate, we applied cosine decays with warm restarts [66]. The Table A in S1 Appendix summarizes all parameters of the network. Results for different parameters are shown in Figs A and B in S1 Appendix. In general, small changes in the network parameters did not affect the results.

Inputs were grayscale images with different shapes (Figs 7 and 8). We added random Gaussian noise with mean $\mu = 0$ and a standard deviation randomly drawn from a uniform distribution. The contrast was varied either by first adding a random value between -0.1 and 0.1 to all pixel values and then multiplying them with a contrast adjustment factor randomly drawn from a uniform distribution or vice versa. The pixel values were clipped between 0 and 1.

### Experiment 1

**Modelling.**   Human data for experiment 1 is based on [28]. We trained CapsNets with the above architecture to solve a vernier offset discrimination task and classify groups of identical shapes. The loss function included a term for shape type classification, a term for vernier offset discrimination, a term for the number of shapes in the image, a term for reconstructing the input based on the network output, and a term for localizing the stimulus position in the image (see Eqs 1–6). The shape repetition, reconstruction and location loss terms are used to stabilize performance but are not essential for producing (un)crowding (see Table A in S1 Appendix). Each loss term was scaled so that none of the terms dominated the others. For the

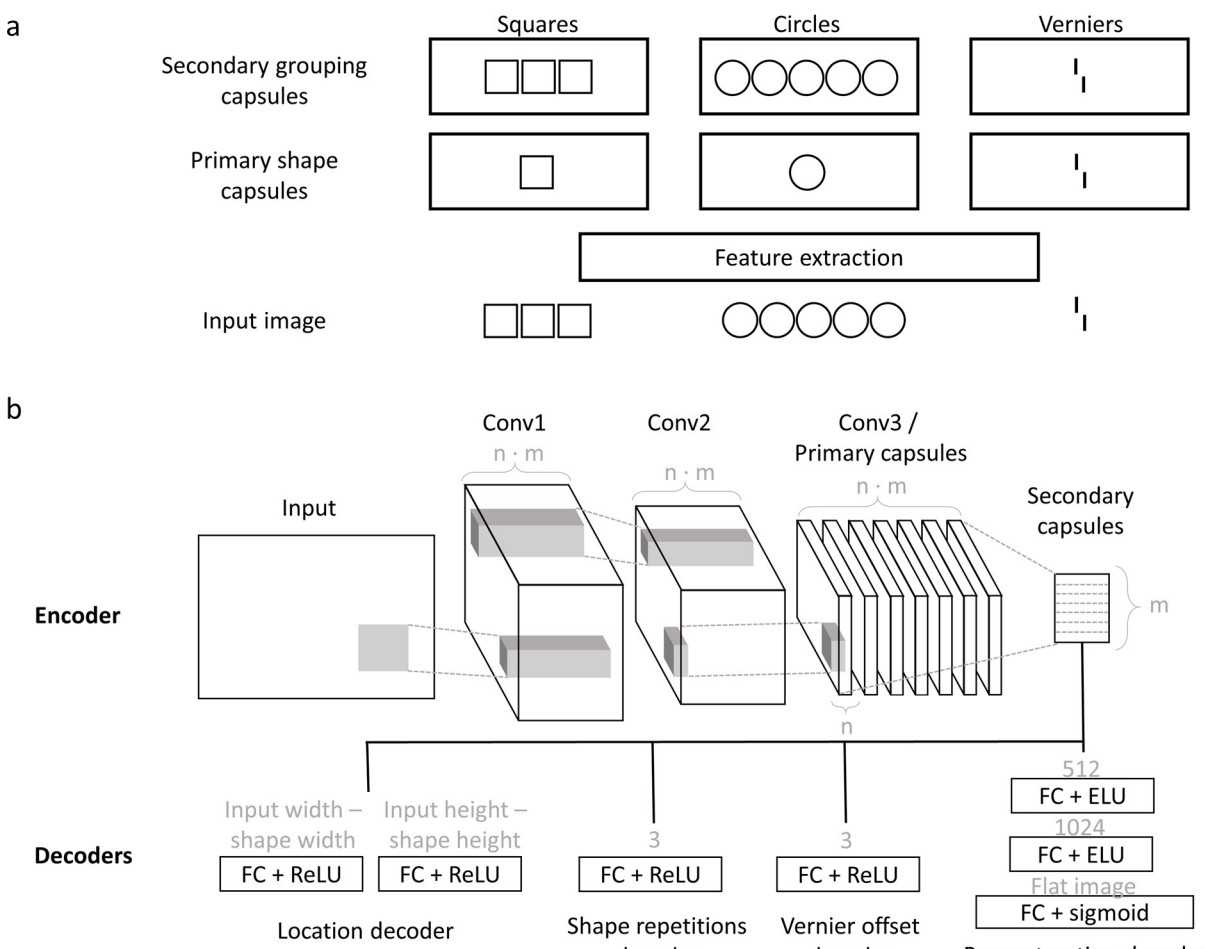

**Fig 6. (a) Ideal representations:** After training, we expected the primary capsules to detect single shapes of different types (here: squares, circles and verniers), and secondary capsules to group these shapes into groups of 1, 3, or 5. If 3 squares are presented, the primary square capsules detect squares at 3 locations. Through routing by agreement, the secondary squares capsule detects this group of 3 squares. If 5 circles are presented, the primary circle capsules detect circles at 5 locations. After routing, the secondary circles capsule represents a group of 5 circles. If a vernier is presented, it is detected and routed to the secondary vernier capsule. **(b) Network architecture:** We used capsule networks with two convolutional layers, one primary capsule layer with m primary capsule types and n primary capsule dimensions, and one secondary capsule layer with m secondary capsule types. The primary capsules were implemented by a third convolution layer, reshaped into m primary capsule types outputting n-dimensional activation vectors (this is the standard way to implement primary capsules). In this example, there are seven primary and secondary capsules types to match the seven shape types used in experiment 1 (see caption a). The primary and secondary capsule layers communicate via routing-by-agreement. Small fully connected decoders are applied to the secondary capsules to compute different loss terms (see main text). The network is trained end-to-end through backpropagation to minimize all losses simultaneously. We used different network hyperparameters in the main text results and in S1 Appendix to ensure that our results are stable against hyperparameter changes (see Table A in S1 Appendix for all hyperparameter values).

shape type classification loss, we implemented the same margin loss as in [39]. This loss enables the detection of multiple objects in the same image. For the vernier offset loss, we used a small decoder to determine vernier offset directions based on the activity of the secondary vernier capsule. The decoder was composed of a single dense hidden layer followed by a ReLU-nonlinearity and a dense readout layer of three nodes corresponding to the labels left and right, or no vernier. The vernier offset loss was computed as the softmax cross entropy between the decoder output and the one-hot-encoded vernier offset labels. The loss term for the number of shapes in the image was implemented similarly, but the output layer comprised three nodes representing the labels one, three or five shape repetitions. For the reconstruction loss, we trained a decoder with two fully connected hidden layers (h1: 512 units, h2: 1024

a

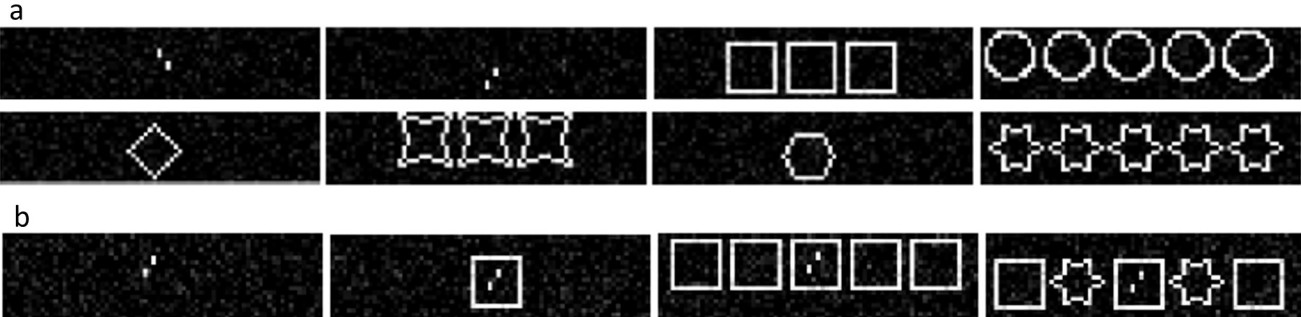

b

**Fig 7. (a) Training stimuli for experiment 1:** All shapes were shown randomly in groups of one, three or five. Verniers were always presented alone. **(b) Testing stimuli for experiment 1:** Example stimuli for the test conditions: In the vernier-alone condition (*left*), we expected the network to perform well on the vernier discrimination task. In crowding conditions (*middle-left*), we expected a deterioration of the discrimination performance. In uncrowding conditions with identical flankers (*middle-right*), we expected a recovery of the performance. In no-uncrowding conditions with alternating flanker types (*right*), we expected crowding.

units) each followed by ELU nonlinearities to reconstruct the input image. The reconstruction loss was calculated as the squared difference between the pixel values of the input image and the reconstructed image. The total loss is given by the following formulas:

$$L_{total} = \alpha_{shape\ type}\ L_{shape\ type} + \alpha_{vernier\ offset} L_{vernier\ offset} + \alpha_{shape\ repetitions}\ L_{shape\ repetitions}$$
$$+\ \alpha_{reconstruction}\ L_{reconstruction} + \alpha_{location}\ L_{location} \quad (1)$$

$$L_{shape\ type} = \sum_k T_k\ \max(0,\ (m^+ - \|v_k\|)^2) + \lambda(1 - T_k)\max(0,\ (\|v_k\| - m^-)^2) \quad (2)$$

$$L_{vernier\ offset} = Xentropy(true\ offset\ direction, predicted\ offset\ direction) \quad (3)$$

$$L_{shape\ repetitions} = Xentropy(true\ shape\ repetition, predicted\ shape\ repetition) \quad (4)$$

$$L_{reconstruction} = \sum_{i,j} (input\ image(i,j) - reconstructed\ image(i,j))^2 \quad (5)$$

$$L_{location} = \sum_{i=x,y} Xentropy(true\ coordinate_i, predicted\ coordinate_i) \quad (6)$$

a

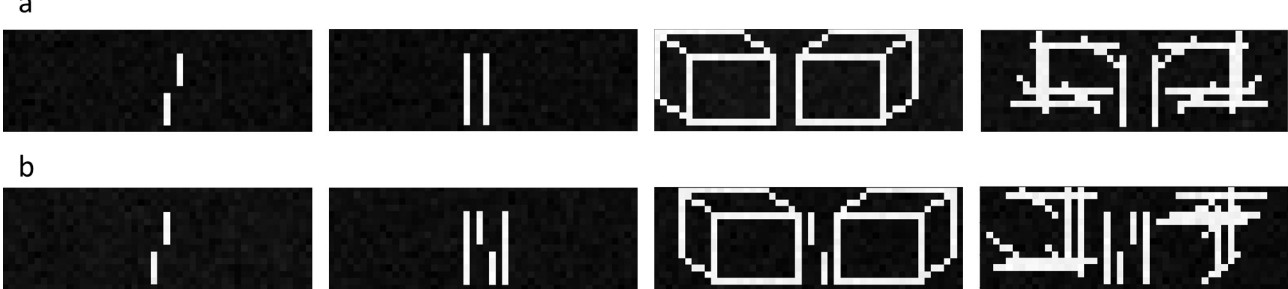

b

**Fig 8. (a) Training stimuli for experiment 2:** Verniers were always presented alone. Lines were presented randomly in groups of 2, 4, 6, or 8. Cuboids and scrambled cuboids were presented in groups of two facing each other. The distance between the shapes varied randomly between 1–6 pixels. **(b) Testing stimuli for experiment 2:** In the vernier-alone condition (*left*), we expected the network to perform well on the vernier discrimination task. In the flanker condition, we expected crowding for all chosen shape types. However, we expect that discrimination performance for the cuboid flankers recovers with increasing routing iterations.

Where the $\alpha$ are real numbers scaling each loss term ($\alpha_{\text{shape type}} = 0.5$, $\alpha_{\text{vernier offset}} = 1$, $\alpha_{\text{shape repetitions}} = 0.4$, $\alpha_{\text{reconstruction}} = 0.0005$, $\alpha_{\text{location}} = 0.1$), $T_k = 1$ if shape class k is present, $\|v_k\|$ is the norm of secondary capsule $k$, and $m^+$, $m^-$ and $\lambda$ are parameters of the margin loss with the same values as described in [39]. All loss terms decreased during training (see S2 Appendix).

The training dataset included vernier stimuli and six different shape types (Fig 7A). Shapes were presented in groups of one, three or five shapes of the same type. The group was roughly centered in the middle of the image. We trained N = 10 networks and averaged their performance. Importantly, the network was never exposed to (un)crowding stimuli during training. Therefore, the network could not trivially learn when to (un)crowd by overfitting on the training dataset. This situation is similar for humans: they know about shapes and verniers, but their visual system has never been trained on (un)crowding stimuli. After training, we tested the vernier discrimination performance on (un)crowding stimuli (Fig 7B), and obtained reconstructions of the input images.

To check that CapsNets explain uncrowding because of the grouping and segmentation capabilities offered by routing by agreement and not merely because of the way they are trained, we replaced the capsule layers by other architectures (a feedforward fully connected layer, a fully connected layer with lateral recurrent connections and a fully connected layer with top-down recurrent connections to the layer below; see Results). All these networks had the same number of neurons as our CapsNets, and we used the same number of recurrent iterations as the number of routing by agreement used for the CapsNets. All networks were trained and tested the same way. The only difference is that CapsNets represent different classes in different capsules, so that we could decode information directly from specific capsules. For example, we could decode vernier offsets specifically from the secondary vernier capsule or reconstruct squares specifically from the secondary squares capsule. The standard ffCNNs with and without recurrent connections do not offer this possibility, because different classes are not represented in different known groups of neurons. Therefore, we decoded vernier offsets, reconstructions, the number of shapes and the shape type from the entire last layer of the network rather than from specific capsules. This difference did not limit the networks' performance, since these architectures performed well during training. Hence, the fact that they do not produce uncrowding is not explained by training limitations, but rather by the fact that they *generalize* to novel inputs differently than CapsNets.

## Experiment 2

**Psychophysical experiment.**   For experiment 2, we collected human psychophysical data. Participants were paid students of the Ecole Polytechnique Fédérale de Lausanne (EPFL). All had normal or corrected-to-normal vision, with a visual acuity of 1.0 (corresponding to 20/20) or better in at least one eye, measured with the Freiburg Visual Acuity Test. Observers were told that they could quit the experiment at any time they wished. Five observers (two females) performed the experiment.

Stimuli were presented on a HP-1332A XY-display equipped with a P11 phosphor and controlled by a PC via a custom-made 16-bit DA interface. Background luminance of the screen was below 1 cd/m$^2$. Luminance of stimuli was 80 cd/m$^2$. Luminance measurements were performed using a Minolta Luminance meter LS-100. The experimental room was dimly illuminated (0.5 lx). Viewing distance was 75 cm.

We determined vernier offset discrimination thresholds for different flanker configurations. The vernier target consisted of two lines that were randomly offset either to the left or right. Observers indicated the offset direction. Stimuli consisted of two vertical 40' (arcmin)

long lines separated by a vertical gap of 4' and presented at an eccentricity of 5° to the right of a fixation cross (6' diameter). Eccentricity refers to the center of the target location. Flanker configurations were centered on the vernier stimulus and were symmetrical in the horizontal dimension. Observers were presented two flanker configurations. In the lines configuration, the vernier was flanked by two vertical lines (84') at 40' from the vernier. In the cuboids configuration, perspective cuboids were presented to the left and to the right of the vernier (width = 58', angle of oblique lines = 135°, length = 23.33'). Cuboids contained the lines from the lines condition as their centermost edge.

Observers were instructed to fixate a fixation cross during the trial. After each response, the screen remained blank for a maximum period of 3 s during which the observer was required to make a response on vernier offset discrimination by pressing one of two push buttons. The screen was blank for 500 ms between response and the next trial.

An adaptive staircase procedure (PEST; [67]) was used to determine the vernier offset for which observers reached 75% correct responses. Thresholds were determined after fitting a cumulative Gaussian to the data using probit and likelihood analyses. In order to avoid large vernier offsets, we restricted the PEST procedure to not exceed 33.3' i.e. twice the starting value of 16.66'. Each condition was presented in separate blocks of 80 trials. All conditions were measured twice (i.e., 160 trials) and randomized individually for each observer. To compensate for possible learning effects, the order of conditions was reversed after each condition had been measured once. Auditory feedback was provided after incorrect or omitted responses.

**Modelling.**   To model the results of experiment 2, we trained CapsNets to solve a vernier offset discrimination task and classify verniers, cuboids, scrambled cuboids and lines. The loss function included the shape type loss, the vernier offset loss and the reconstruction loss from experiment 1 (Eqs 2, 3 and 5):

$$L_{total} = \alpha_{shape\ type} L_{shape\ type} + \alpha_{vernier\ offset} L_{vernier\ offset} + \alpha_{reconstruction} L_{reconstruction} \tag{7}$$

The training dataset included vernier stimuli and one of three different shape types (lines, cuboids, scrambled cuboids; Fig 8A). The scrambled cuboids were included to make the task harder, and to prevent the network from classifying cuboids simply based on the number of pixels in the image. Similarly, the lines were randomly presented in a group of 2, 4 or 6 to prevent the network from classifying cuboids simply based on the number of pixels in the image. Both, cuboids and shuffled cuboids were always presented in groups of two facing each other. The distance between the shapes was varied randomly between one to six pixels. As before, (un)crowding stimuli were never presented during training. Therefore, the network could not trivially learn when to (un)crowd by overfitting on the training dataset.

After training, we tested the vernier discrimination performance for (un)crowding stimuli (Fig 8B) while varying the number of recurrent routing by agreement iterations from 1 to 8 iterations. We trained 50 networks with the same hyperparameters and averaged their performance. Some networks were excluded because vernier discrimination performance for *both* line and cuboid flankers was at ceiling ($> = 95\%$) or floor ($< = 55\%$). Qualitatively, these exclusions do not change our results. However, the fact that a few networks are at floor or ceiling is misleading.

## Supporting information

**S1 Appendix. Results are robust against hyperparameter changes.**
(PDF)

**S2 Appendix. Loss evolution during training.**
(PDF)

**S3 Appendix. Performance deteriorates due to crowding.**
(PDF)

## Author Contributions

**Conceptualization:** Adrien Doerig, Michael H. Herzog.

**Data curation:** Adrien Doerig, Lynn Schmittwilken, Bilge Sayim, Mauro Manassi.

**Formal analysis:** Adrien Doerig, Lynn Schmittwilken.

**Funding acquisition:** Michael H. Herzog.

**Investigation:** Adrien Doerig, Lynn Schmittwilken, Bilge Sayim, Mauro Manassi.

**Methodology:** Adrien Doerig, Lynn Schmittwilken, Bilge Sayim, Mauro Manassi, Michael H. Herzog.

**Software:** Adrien Doerig, Lynn Schmittwilken.

**Visualization:** Adrien Doerig, Lynn Schmittwilken.

**Writing – original draft:** Adrien Doerig, Michael H. Herzog.

**Writing – review & editing:** Adrien Doerig, Lynn Schmittwilken, Michael H. Herzog.

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
