## [Decision Letter · Decision Letter 0]

27 Feb 2020

Dear Mr. Doerig,

Thank you very much for submitting your manuscript "Capsule Networks as Recurrent Models of Grouping and Segmentation" for consideration at PLOS Computational Biology.

As with all papers reviewed by the journal, your manuscript was reviewed by members of the editorial board and by several independent reviewers. In light of the reviews (below this email), we would like to invite the resubmission of a significantly-revised version that takes into account the reviewers' comments.

In addition to the technical comments by the reviewers, I would like to ask you to carefully clarify why this manuscript constitutes a major advance over your previous papers in PLoS Computational Biology and elsewhere. If we figure the advance to be too incremental for PLoS Computational Biology, accepting the paper to PLoS One instead might be a potential suggestion resulting from the re-review process. (Of course, this does not make any statement on the probability of this to be the result, just making you aware in advance that this option exists for papers that are found to be technically correct, but are found to not provide a sufficent advancement over previous knowledge, in this case authors can be offered to get their paper accepted as is or with minor revisions PLoS One; obviously it remains the decision of the authors as to whether they want to accept the offer or not). 

We cannot make any decision about publication until we have seen the revised manuscript and your response to the reviewers' comments. Your revised manuscript is also likely to be sent to reviewers for further evaluation.

Sincerely,

Wolfgang Einhäuser

Deputy Editor

PLOS Computational Biology

Reviewer's Responses to Questions

**Comments to the Authors:**

Reviewer #1: I have enjoyed reading this paper. The authors are probing the usability of capsule networks on a task they are supposed to (and actually do) solve, spatial configurations, with, as an emergent feature, behavior (mostly) matching (published) human perception on the occurrence of crowding and uncrowding.

To link processing in capsule networks to the type of computations involved in recurrent processing the authors conduct a second series of experiments. When humans are presented with a masking stimulus that is presumed to require recurrent processing to resolve, uncrowding emerges for a longer stimulus duration. This also occurs in capsule networks with more routing iterations.

This is a timely paper extending the range of AI networks being used to probe cognition. Also, experiment 2 has the potential of becoming an instant citation classic (I will certainly also include it in lectures to illustrate the types of experiments you can do with DNNs).

Major points

I a missing a lot of details on the network and have encountered some inconsistencies.

• Can you indicate, also in the figure, how the loss is propagated from the loss to the output? Specifically, how are the different sources of loss inserted into the output layer.

• Caps of fig 5a and L373 refer to 3 convolutional layers on the text, the graphic, and main text to two layers. I suspect the latter. What are the dimensions of the input images? What are the specs of the convolutional layers? M refers to number of shape types. What is N exactly? How is it determined?

• What are the other parameters of the conv layers?

• If there are 3 conv layers this would be somewhat remarkable given the input images. Could you provide more details on why 3 are needed?

• Fig 5a refers to secondary capsules, I suspect these are the output capsules (but not the output layer).

• Please have fig 5a include the entire network including, including the output layer (and how it relates to the different loss functions).

• L373 states there are dropout layers between the thee convolutional layers. I suspect this is not possible.

• What is the development of loss during training? How is the loss of the other networks?

The results from the psychophysical experiment appear to be brilliant, but only two lines. It would be even more convincing / impactful with an additional point. What would a Necker cube do? What would a square do?

• Could you speculate further on how you see the relationship between capsule networks, recurrent processing and the brain. Do the authors think that capsule networks are a way (of many) in which grouping can be solved? L351-L366 shows signs of redaction after earlier reviewer comments but ends up at: things could be something. I realize different reviewers have different comments but I would invite the authors to take more of a position.

To mention

• The experiment has been programmed in an early version of Tensorflow, this will be more difficult to track once the paper is published. Would it be possible to update this code to Tensorflow 2.0, or alternatively, the latest version of Tensorflow 1.x?

Minor points

• L53, apart from grouping another major difference, relevant to mention, is that DNNs make different mistakes than humans in classification.

• The authors might want to point to the developing literature on capsule networks in tumor detection, both highlighting its general use, and its underexposure in CCN.

• Capsule networks allow you to decode specific information from specific capsules. You can read this as a hack in simulating recurrent processing but also as a feature of higher-tier areas when dealing with lower-tier common input. You could have more shine on the capsule networks here (and this, as far as I can see, really adds to the current corpus of D(R)NNs.

Reviewer #2: I found this paper very interesting and, overall, I don't think it requires much additional work/revision. From my viewpoint, there is only one major concern, and it is overlap with prior publications from the same group. For example, in a previous publication in the same journal last year (Doerig et al 2019), the authors make similar points, in particular the fact that feedforward architectures like cnn do not explain uncrowding. The manuscript under revision does add to that prior publication, but it is more of an incremental addition than an entirely new concept. As far as I can see, the main contribution is to take an established neural network architecture like capsule nets, and let the capsules do the job that was formerly carried out by LAMINART. It is not quite that simple, and of course capsule nets is an attractive integrated approach rather than a hybrid one, I get that, but if you look at the bigger picture that's more or less correct. I don't think there is a general problem with this approach i.e. that of publishing many papers all inching one step at a time towards a final goal. I personally don't subscribe to this approach, but I understand that it is common in the field and so the authors should not be penalized for conforming to it like many other groups. In the end, it is for editors to judge whether the incremental knowledge is sufficient to justify another publication in PLOS CB or not. As far as I am concerned in my role as reviewer, when I evaluate this paper in and of itself (without comparing it to others by the same group), I think the paper is good and interesting.

There are potentially several specific aspects that one could pick upon and demand endless variants of the trainign protocol, inclusion of other architectures, and so on. But I don't think that is necessarily productive, because the authors have already done a good job of exploring different aspects and variants of this topic. One obvious issue, which the authors identify as being unresolved, is the separation of crowding stimuli from the vernier target during training. It is clear why the authors used this approach (i.e. to steer the network towards grouping the crowding array as being a separate element than the vernier offset), and it seems sensible. It would be nice if the network could figure that out for itself, but I guess that will be for another paper.

**Have all data underlying the figures and results presented in the manuscript been provided?**

Reviewer #1: No: • The authors state they will release the models and code on github on publication. I cannot evaluate what this means exactly, and how extensive this is.

• The behavioral data from experiment 2 is not mentioned as data to release, and has not been released.

Reviewer #2: None

PLOS authors have the option to publish the peer review history of their article (what does this mean?). If published, this will include your full peer review and any attached files.

Reviewer #1: Yes: H.Steven Scholte

Reviewer #2: No
---

## [Decision Letter · Decision Letter 1]

4 Jun 2020

Dear Mr. Doerig,

We are pleased to inform you that your manuscript 'Capsule Networks as Recurrent Models of Grouping and Segmentation' has been provisionally accepted for publication in PLOS Computational Biology.

Best regards,

Wolfgang Einhäuser

Deputy Editor

PLOS Computational Biology

Wolfgang Einhäuser

Deputy Editor

PLOS Computational Biology

Reviewer's Responses to Questions

**Comments to the Authors:**

Reviewer #1: I want to thank the authors for addressing my points to extensively. They have put a lot of effort into them, and for me at least, it is now much clearer what the architecture of the network is exactly, and also what is difficult for the network to learn.

Also, the authors now provide the complete model on github. And fair enough to keep it in the version it was used (and probably better also).

Finally, with regards to minor comment: "• L53, apart from grouping another major difference, relevant to mention, is that DNNs make different mistakes than humans in classification", you are of course right. This is exactly what you discuss in the section.

I think this is a very relevant paper, with the second part expanding the earlier paper of by these authors, and the second part substantially adding to our knowledge with a novel approach that compares human behavior with evolving performance of the capsule network.

Reviewer #2: I am happy with the revision. Just to be clear (something I have already pointed out in the previous round), I did not mean to say there was nothing new in this manuscript. I meant to say that there was substantial overlap with previous work by the same group, which I believe is a fair account. This does not mean that there is nothing new.

A small piece of advice for future revisions: please make sure that you clearly mark all changes you have applied to the manuscript e.g. by using colored text or other, so that reviewers can easily identify what was added/modified. It does save time to reviewers (and ultimately to yourselves).

**Have all data underlying the figures and results presented in the manuscript been provided?**

Reviewer #1: Yes

Reviewer #2: None

PLOS authors have the option to publish the peer review history of their article (what does this mean?). If published, this will include your full peer review and any attached files.

Reviewer #1: Yes: H S Scholte

Reviewer #2: No

---

## [Editor Report · Acceptance letter]

13 Jul 2020

PCOMPBIOL-D-19-02035R1 

Capsule Networks as Recurrent Models of Grouping and Segmentation

Dear Dr Doerig,

I am pleased to inform you that your manuscript has been formally accepted for publication in PLOS Computational Biology. Your manuscript is now with our production department and you will be notified of the publication date in due course.

With kind regards,

Sarah Hammond
